# Training neural networks faster with minimal tuning using pre-computed lists of hyperparameters for NAdamW

## Abstract

If we want to train a neural network using any of the most popular optimization algorithms, we are immediately faced with a dilemma: how to set the various optimization and regularization hyperparameters? When computational resources are abundant, there are a variety of methods for finding good hyperparameter settings, but when resources are limited the only realistic choices are using standard default values of uncertain quality and provenance, or tuning only a couple of the most important hyperparameters via extremely limited hand-designed sweeps. Extending the idea of default settings to a modest tuning budget, Metz et al. (2020) proposed using ordered lists of well-performing hyperparameter settings, derived from a broad hyperparameter search on a large library of training workloads. However, to date, no practical and performant hyperparameter lists that generalize to representative deep learning workloads have been demonstrated. In this paper, we present hyperparameter lists for NAdamW derived from extensive experiments on the realistic workloads in the ALGOPERF: Training Algorithms benchmark. Our hyperparameter lists also include values for basic regularization techniques (i.e. weight decay, label smoothing, and dropout). In particular, our best NAdamW hyperparameter list performs well on ALGOPERF held-out workloads not used to construct it, and represents a compelling turn-key approach to tuning when restricted to five or fewer trials. It also outperforms basic learning rate/weight decay sweeps and an off-the-shelf Bayesian optimization tool when restricted to the same budget.

## 1 Introduction

In order to train a neural network, practitioners need to choose a training algorithm and the associated hyperparameter values. This procedure generally involves large scale tuning studies requiring lots of computational resources to arrive at the best performing hyperparameters. In resource-constrained settings, such tuning protocols lead to poor hyperparameter choices, as do protocols based on limited, low-dimensional sweeps or library defaults. In some cases, we can only afford a handful of tuning trials. Such a constraint can occur both when computational resources are limited in an absolute sense, and when they are abundant, but the cost of a single training run is also quite large. The latter situation requires deriving scaling ladders based on smaller versions of the same problem and having access to good defaults at lower scales is helpful to practitioners in this situation.

For concreteness, let us assume we are restricted to a modest tuning budget of 5 trials, run in parallel,[1] and must select a hyperparameter configuration for a new training problem. There are an endless variety of possible tuning procedures we could use. For example, we could fix all other hyperparameters to default values and just sweep the learning rate over 5 different values spaced uniformly on a log scale. We could use a random search on a small, handcrafted search space. We could handpick 5 specific points based on previous experience with other problems and any intuition we might have. We could build a smaller version of the training problem, do more extensive tuning studies, and then apply scaling heuristics (e.g. using $\mu$-parameterization (Yang et al., 2022)) to transfer the settings back to the original problem. We could even

---

[1]Although arbitrary, a budget of 5 parallel trials matches the external tuning rules of the ALGOPERF training algorithms benchmark (Dahl et al., 2023), which we make extensive use of in the rest of this work.

run a Bayesian optimization tool, such as Vizier (Song et al., 2023), on a handcrafted search space, although technically if we ran 5 *sequential* trials that would be a more generous budget than our assumption of 5 trials that ran in parallel. At the end of the day, we need to have some procedure that works well and is reproducible, so it behooves us to try and measure the efficacy of different natural protocols.

In this paper, we study the problem of finding a good hyperparameter configuration on a new problem in only a handful of parallel tuning trials. We draw inspiration from Metz et al. (2020), who suggested a tuning protocol using precomputed, ordered lists of hyperparameter configurations. The essence of their hyperparameter-list approach is to find a list of configurations that do well on a diverse library of training workloads and then try them, in priority order, on a new problem, up to the limit of the tuning budget. By defining an ordered list, Metz et al. (2020) have a concise way of recommending a set of configurations for any budget up to the size of their full list. Unfortunately, the specific precomputed hyperparameter lists offered by Metz et al. (2020) do not perform as well as we would like on typical deep learning benchmark workloads, and do not train successfully on any of the workloads we tried, and thus are unlikely to be very useful to practitioners. To the best of our knowledge, to date there has been no convincing evidence that such an ordered list of hyperparameter configurations can outperform simple alternative tuning procedures on typical deep learning workloads that weren't used to build the lists, even at modest tuning budgets. In this work, we show that, for a restrictive tuning budget of 5 parallel trials, our hyperparameter lists outperform alternative tuning protocols in leave-one-workload-out cross validation on the workloads of the AlgoPerf: Training Algorithms benchmark (Dahl et al., 2023), as well as performing well on held-out variant workloads. We chose to construct hyperparameter lists for a variant of Adam (Kingma & Ba, 2014): Nesterov Adam with decoupled weight decay (NAdamW) (Dozat, 2016; Loshchilov & Hutter, 2019). We elected NAdamW because it outperformed AdamW in Dahl et al. (2023), as well as in our own anecdotal experience. Specifically, in this work we make the following contributions:

1. Similar to Metz et al. (2020), we articulate a simple and generic method that produces hyperparameter lists given a (potentially broad) hyperparameter search space and a library of workloads.

2. We show that a 5-point hyperparameter list for NAdamW constructed by our method can successfully train 7 of the 8 AlgoPerf base workloads, measured with leave-one-out cross validation over workloads, and that our best 5-point hyperparameter list generated using all base workloads also successfully trains on official AlgoPerf workload variants designed to measure training algorithm robustness to small architectural changes.

3. We show that our NAdamW hyperparameter lists (once again measured using leave-one-out cross validation over the AlgoPerf base workloads) outperform other natural alternative procedures for resource-constrained hyperparameter tuning that are also restricted to the same budget of a modest number of parallel trials (and also don't have access to any special information about the workload). Specifically, compared to quasi-random search on the broad hyperparameter search space we used to build our lists, on most workloads our lists are more than three times more efficient in tuning.

Collectively, our results show that—perhaps surprisingly—at least for an Adam-style training algorithm at a limited tuning budget, we can find a small set of hyperparameter configurations that generalize well enough to new workloads to be competitive with other alternative tuning procedures. Despite the importance of workload-specific hyperparameter tuning in deep learning, NAdamW combined with our hyperparameter list becomes a competitive *fully-specified* training algorithm requiring no user intervention whatsoever to apply. Although there are a few caveats to our results that we discuss in Section 5, we believe that our work is an important step on the path towards training algorithms that are easier to use and away from under-specified algorithms with free parameters that both need to be tuned per-workload and don't include guidance on how to tune them (as a function of the budget). We view our work as mostly orthogonal to research on scaling heuristics that attempts to predict good hyperparameter configurations for larger models based on configurations that work well for smaller models. In our case, we do not assume access to a "scaling ladder" of increasingly expensive versions of the same workload where we could afford to greatly exceed our budget of tuning trials.

## 1.1 Related Work

Hyperparameter-optimization has a long history of research spanning different disciplines beyond machine learning. Several works in literature have highlighted the need for clear accounting for this crucial and resource-intensive but seldom-reported part of machine learning pipelines (Sculley et al., 2018; Hutter et al., 2019; Eggensperger et al., 2019). While a detailed description of this broad research area is out of the scope of the paper, in this section we provide a a short survey of some of the relevant techniques in the context of machine learning.

Perhaps the most straighforward approach to selecting hyperparameters is via a grid/manual search. Bergstra & Bengio (2012) show that random search outperforms grid/manual search, especially where there are a few critical hyperparameters. Bousquet et al. (2017) propose the use of quasi-random search via low discrepancy sequences instead of random search and show that these are more efficient in the context of deep-learning models. Several alternatives to grid/random/quasi-random search have been studied in literature across various disciplines and can broadly be categorized as follows, Bayesian optimization (Bergstra et al., 2011; Snoek et al., 2012; Shahriari et al., 2015), Derivative-free optimization (Conn et al., 2009; Rios & Sahinidis, 2013), as an instance of the Multi-armed bandits problem (Ginebra & Clayton, 1995; Srinivas et al., 2009; Jamieson & Talwalkar, 2016; Li et al., 2018), efficient learning of Decision Trees (Hazan et al., 2017), scaling laws (Kadra et al., 2023) or combinations of these ideas (Falkner et al., 2018).

Metz et al. (2021) introduced the idea of ordered lists of hyperparameters derived from a training runs of small-scale machine learning problems, however these hyperparameter lists were not generated using training runs of problems that are representative of the typical deep learning practitioner and so do not perform well on real world optimization tasks. To the best of our knowledge a hyperparameter list that demonstrates good performance on medium to large scale deep learning tasks is missing from the literature.

Evaluation of hyprameter-optimization algorithms necessitates a standardized method to benchmark performance of the training algorithm itself. While various proposals exist in literature, e.g. Schneider et al. (2019); Moreau et al. (2022), for our purpose we choose the recently proposed ALGOPERF (Dahl et al., 2023) benchmark. ALGOPERF introduces a time-to-result training algorithms benchmark on fixed hardware on realistic deep learning workloads comprising of a wide variety of datasets and model architectures relevant to practitioners. We develop our hyperparameter-list on ALGOPERF base workloads and test the robustness of the final hyperparameter lists on ALGOPERF heldout workloads. ALGOPERF heldout workloads are designed such that the hyperparameters that work well for base workloads are not necessarily the best choices for heldout variants of the corresponding base workload. Thus developing a hyperparameter list that reaches targets on both base and held-out workloads in competitive times is both challenging and meaningful to practitioners as it's a strong indicator of generalization to a wide variety of problems.

## 2 Methods

Our goal in this work is to produce ordered hyperparameter lists that generalize to previously unseen problems. Given a unified search space specifying search ranges for each hyperparameter in a problem agnostic manner as input, our method produces an ordered hyperparameter list of a desired size. The method first samples hyperparameter points from the input search space to generate possible candidates for the final hyperparameter list. These hyperparameter points are then used to train on a library of predetermined workloads. Using these trials, we use a greedy procedure to incrementally construct the final hyperparameter list, using a simple cost function to rank the possible points to include at each step. To evaluate the quality of the final list of points, we use a heldout set of workloads as well as leave-one-out cross validation at the workload level. All of our experiments focus on finding hyperparameters for NAdamW, although any update rule accompanied by a reasonable broad search space could be used just as easily.

### 2.1 Training and Update Rule Details

Several aspects of the training pipeline were fixed and not tuned. We used a learning rate schedule comprised of a linear warm-up segment followed by cosine decay to 0 in all our experiments. We used the NadamW

update rule as implemented in Optax (DeepMind et al., 2020).[2] All our experiments ran on TPUv2 2x2 slices (16GB HBM per chip), except the CRITEO 1TB workload which was trained on TPUv3 4x4 (32GB HBM per chip) due to higher memory requirements in the embedding layers. This hardware choice departs from the ALGOPERF benchmark which uses 8x Nvidia V100 GPUs for training runs.

## 2.2  Broad Search Space

Table 1 shows the broad search space used to sample candidate points for hyperparameter point lists. The search space is designed to be broad enough that, for most workloads, it should include some points that perform well. Unlike default settings in libraries that implement various optimizers (e.g. Adam), our search space includes several regularization hyperparameters, namely weight decay, dropout, and label smoothing (results in Dahl et al. (2023) indicate they can be useful on at least some AlgoPerf workloads and they are all popular on many other workloads as well). One potential issue with using default settings for, say, Adam is that overfitting can happen quite easily on some workloads. For a list of hyperparameter points to have any hope of generalizing across disparate learning problems, the initial search space must not completely preclude sampling points that enable some kind of regularization. It is especially important to consider the weight decay strength jointly with optimization hyperparameters, such as the learning rate, since in practice they can interact for many optimizers (and the exact interaction depends on the details of the update rule parameterization).

Since learning rate warmup can improve training stability at higher peak learning rates (Gilmer et al., 2022) but too long a warmup phase might slow down training, we tuned the percentage of training used for learning rate warmup between 2%, 5% and 10%. For simplicity, we used a schedule that decays the learning rate all the way to zero at the end of training. For the $\beta_1$ and $\beta_2$ hyperparameters, the defaults in the Optax (DeepMind et al., 2020) AdamW implementation we used are 0.9 and 0.999, so we constructed a search range that allows for points around those values. Specifically, we tuned $1 - \beta_1$ and $1 - \beta_2$ on a log scale between 1e-3 and 0.2. The higher end of the range corresponds to $\beta$ values of 0.8 which ensure that the best hyperparameters found in Algoperf target setting experiments are contained within our search range. We used a discrete set of values to search for dropout probability, and used the same dropout probability for any and all dropout layers in the workload. We sampled 200 hyperparameter points from our broad search space using quasirandom search (Bousquet et al., 2017) and reused these points across all ALGOPERF base workloads, resulting in 8 trials per hyperparameter point (one for each workload).

| Hyperparameter | Range | Scale |
|---|---|---|
| Base LR | [1e-4, 1e-2] | Log |
| $1 - \beta_1$ | [1e-3, 0.2] | Log |
| $1 - \beta_2$ | [1e-3, 0.2] | Log |
| Warmup | $\{2\%, 5\%, 10\%\}$ | Discrete |
| Schedule | linear warmup + cosine decay | |
| Weight Decay | [1e-4, 0.5] | Log |
| Label Smoothing | $\{0.0, 0.1, 0.2\}$ | Discrete |
| Dropout | $\{0.0, 0.1\}$ | Discrete |

Table 1: Broad search space used for NadamW

## 2.3  AlgoPerf Base Workloads

Our procedure to generate hyperparameter lists depends on selecting lists that perform well across a library of training workloads, with a workload consisting of a dataset, model architecture, loss function, and (potentially) an evaluation metric. Specifically, in this work we made use of the various workloads of the AlgoPerf benchmark (see Table 2). In all our experiments, we used workload implementations from the init2winit (Gilmer et al., 2023) github repository. Note that some minor differences exist between the init2winit imple-

---

[2]When using the hyperparameter lists we found, users should follow the Optax implementation for the update rule closely since optimizer update rule implementations can have subtle, but important, differences across frameworks and libraries implementing them.

mentations of the various AlgoPerf workloads and the AlgoPerf paper competition implementations.[3] See Appendix A for additional implementation details. The set of workloads included in the AlgoPerf benchmark can all be trained on a single machine with multiple accelerators attached and represent a wide variety of model architectures, datasets, and modalities.

| Task | Dataset | Model | Loss | Metric | Validation Target |
|------|---------|-------|------|--------|-------------------|
| Clickthrough rate prediction | Criteo 1TB | DLRMsmall | CE | CE | 0.123649 |
| MRI reconstruction | fastMRI | U-Net | L1 | SSIM | 0.723653 |
| Image classification | ImageNet | ResNet-50
ViT | CE
CE | ER
ER | 0.22569
0.22691 |
| Speech recognition | LibriSpeech | Conformer
DeepSpeech | CTC
CTC | WER
WER | 0.078477
0.1162 |
| Molecular property prediction | OGBG | GNN | CE | mAP | 0.28098 |
| Translation | WMT | Transformer | CE | BLEU | 30.8491 |

Table 2: **Summary of the AlgoPerf base workloads reproduced from Dahl et al.** The possible losses are the cross-entropy loss (CE), the mean absolute error (L1), and the Connectionist Temporal Classification loss (CTC). The evaluation metrics additionally include the structural similarity index measure (SSIM), the error rate (ER), the word error rate (WER), the mean average precision (mAP), and the bilingual evaluation understudy score (BLEU).

### 2.3.1 Significance of AlgoPerf workload targets

Dahl et al. (2023) provides validation error targets for each of the AlgoPerf workloads and defines successful training as achieving these targets. We also adopt this definition of successful training, since the AlgoPerf targets were set using extensive tuning experiments using multiple training algorithms. Reaching these targets quickly is non-trivial and typically requires extensive hyperparameter tuning (including regularization hyperparameters), along with careful choice of training algorithms. AlgoPerf also specifies validation set targets for it's workload variants. The workload variants were specifically designed such that hyperparameters that train successfully on the corresponding base workloads, using Adam, would likely not be the best choice on the variants workloads, although in hindsight some variants depart more from the workload they are based on than others. Thus if a hyperparameter point that performs well on a particular base workload can't reach the goal on a variant, then it is unlikely to do well on unseen workloads.

AlgoPerf specifies targets on validation set evaluation metrics instead of validation loss, in order to be closer to the objectives of the applied problems the workloads are derived from. Metz et al. (2020) priorities the training loss in their hyperparameter list construction, but given that we ultimately evaluate a model using it's validation set performance, we stick with the AlgoPerf convention of validation set targets to evaluate our hyperparameter lists. Metz et al. (2020) doesn't use any notion of fixed targets on loss or validation metrics which is an important distinction from the AlgoPerf benchmark used in our work.

### 2.4 AlgoPerf Workload Variants

In addition to leave-one-out experiments on the AlgoPerf base workloads, we also used a subset of the AlgoPerf workload variants to evaluate the performance of our final hyperparameter lists on unseen problems. Dahl et al. (2023) constructed variants of the base workloads by introducing architectural modifications intended to make the best hyperparameters differ between the base workloads and the corresponding variants. We selected the variant for each base workload that seemed most challenging (the variant where the best hyperparameter point for the base workload had the lowest ranking among points scored on the variant).

In order for a hyperparameter list to plausibly perform well on unseen, novel problems, it should at least train successfully on variations of the workloads it was developed on. Table 3 details the variants chosen (from

---

[3]The Criteo1TB dataset for our experiments comes from an older, but now unavailable, version so results might have slight mismatch from AlgoPerf repository https://github.com/mlcommons/algorithmic-efficiency.

the pool described in Dahl et al. (2023) ) for our experiments, along with the architectural modifications applied to the corresponding base workloads to produce them.

| Base Workload | Variant | Variant Description | Validation Target |
|---|---|---|---|
| **Criteo 1TB** | | | |
| DLRMSMALL | LayerNorm | Adds layer normalization to the network | 0.123757 |
| **fastMRI** | | | |
| U-NET | LayerNorm | Replaces instance normalization with layer normalization with learnable parameters | 0.723284 |
| **ImageNet** | | | |
| RESNET-50 | SILU | Replaces all ReLU activations with SiLU | 0.24555 |
| VIT | POST-LN | Uses POST-LN instead of PRE-LN | 0.246880 |
| **LibriSpeech** | | | |
| CONFORMER | GELU | Replaces all ReLU activations with GELU | 0.094114 |
| DEEPSPEECH | TANH | Replaces all ReLU activations with TanH | 0.150883 |
| **OGBG** | | | |
| GNN | GELU | Replaces all ReLU activations with GELU | 0.27771 |
| **WMT** | | | |
| TRANSFORMER | POST-LN | Uses POST-LN instead of PRE-LN | 30.0779 |

Table 3: Description of ALGOPERF workload variants selected from Dahl et al. (2023) that were used for evaluating hyperparameter lists in terms of robustness to architectural modifications.

## 2.5 Cost function for evaluating a hyperparameter list

There are many different ways to define a cost function to evaluate hyperparameter lists, including several natural choices from previous work. For instance we can view the benchmark score used in AlgoPerf itself as an implicit cost function: Dahl et al. (2023) used performance profiles (Dolan & Moré, 2002) to compare different hyperparameter lists, but these are relative scores dependent on the pool of competing lists being compared. Dahl et al. (2023) also proposed an absolute score by taking the geometric mean of per-workload training times (time to achieve the workload's validation error goal) as a measure to indicate year-over-year progress in the ALGOPERF benchmark. These absolute scores are much more suitable for our purposes of comparing thousands of candidate hyperparameter lists, since our particular pool of lists isn't inherently meaningful and is only an artifact of our procedure for finding a well-performing list. We built upon the idea of taking a geometric mean over training times on individual workloads to design a cost function that takes as input a candidate hyperparameter list, along with the associated training trials, and produces an absolute score signifying how well it does on the ALGOPERF benchmark workloads. Given a candidate list of hyperparameters, we first define per-workload step fractions, as these form the building blocks of our cost function. The step fraction for a given workload is defined as the ratio of the smallest step at which the target is achieved, across all hyperparameter points in the list, to the maximum number of steps allowed for that workload. If a particular workload target is not achieved by any hyperparamter point in the list within the stipulated time budget, we assign a fixed score given by a parameter $\tau > 1$ referred to as a *penalty factor*.

More precisely, let $N$ be the total number of workloads and $L$ be an input hyperparameter list of $K$ points being evaluated. Our cost function receives the hyperparameter list as input along with the associated trials array of size $K$x$N$ corresponding to $K$ trials per workload [4]. For the $i^{th}$ workload, let $T_i$ be total step budget, i.e. the maximum number of steps a trial was run for, for that workload. We define $t_{ki}$ as the first step in the $k^{th}$ trial to hit the validation target for $i^{th}$ workload and if target is not reached, then $t_{ki} = \infty$. We further define $t_i$ as:

$$t_i = min(t_{1i}, t_{2i}, ..., t_{ki}, ..., t_{Ki}).$$

For $N$ workloads and a candidate hyperparameter list $L$, our cost function is defined as :

$$\mathcal{C}_\tau(L) = \prod_{i=1}^{N} \min\left(\frac{t_i}{T_i}, \tau\right)^{1/N}$$

---

[4]We suppress the latter input from our notation as it is clear from the context

Our cost function is designed such that it assigns hyperparameter lists that achieve a larger number of targets, faster a lower score. The value of $\tau$ naturally affects the choice of the optimal hyperparameter list, with larger penalties rewarding lists that can train on a larger fraction of workloads successfully. See Appendix C for an ablation test for the values for $\tau$ on a leave-one-workload-out experiment. From the ablation experiment, we found that increasing the penalty factor above its minimum of 1.0 results in hyperparameter lists that perform better on unseen workloads. We eventually choose a value of 2.0, as the choice of the optimal hyperparameter list seems stable around this value.

Our cost function differs from Metz et al. (2020). They use a cost function defined using a sum over the best achieved (normalized) loss values across different workloads. Within each workload, they normalize the entire training curve such that the loss values fall in the range of [0, 1]. They perform this normalization by mapping the loss achieved with the random initial weights to 1 and mapping the best achieved loss value at any point in training across any hyperparameter configuration to 0. If a loss value greater than 1 is encountered, it is clipped to 1. They take the mean over normalized loss values from the entire training curve (i.e. they the loss of different iterates throughout training) and feed it into the cost function. An immediate concern with such normalization strategiesis that they might not work well in practice since different workloads have different loss functions and thus different natural loss scales. By operating on training times in our cost function, it receives values with the same units (time) for all workloads, avoiding the need to normalize loss values across disparate tasks and replacing it with a need for loss thresholds, per-workload, that define successful training, which we can reuse from the AlgoPerf benchmark.

## 2.6 Producing the final hyperparameter list

Now that we have described a cost function to score different candidate lists using measurements of how fast they trained on a library of workloads, we can describe how the final list is selected. More precisely, given $P$ hyperparameter points along with the associated training runs (trials) on a set of $N$ workloads, to produce a final list of K points, in order, we use the greedy procedure below:

---
**Algorithm 1** Procedure to greedily select the best hyperparameter points list of length K.

---
$P \leftarrow$ set of all candidate hyperparameter points drawn from broad search space $S$
$N \leftarrow$ number of workloads used to develop hyperparameter list
$\mathcal{C}_\tau \leftarrow$ cost function
$i \leftarrow 0$
$L \leftarrow []$   (list of hyperparameter points to be returned)

**while** $i \neq K$ **do**
    $p^* = \arg\min_p \mathcal{C}_\tau(L + p)$   $\forall p \in P$ and $p \notin L$              ▷  $L + p$ denotes the concatenation of point $p$ to candidate list $L$
    $L = L + p^*$
    $i = i + 1$
**end while**

---

At each step of our greedy procedure, until reaching a list of $K$ points, we add the hyperparameter point that most reduces the cost of the overall hyperparameter list formed so far.

We also considered a simple exhaustive strategy to pick a *set* of hyperparameter points of size $K$, instead of an ordered list. The exhaustive procedure evaluates all possible $\binom{P}{K}$ hyperparameter set candidates and selects the one with the lowest cost. However, the exhaustive strategy does not produce ordered hyperparameter lists that conveniently express a list for all smaller budgets less than $K$. In other words, given the best hyperparameter list of size $K$, by virtue of being ordered, the first $K - 1$ represent a list of size $K - 1$. We choose the greedy strategy both because it produces ordered hyperparameter lists that provide guidance for a range of tuning budgets and because the greedy procedure yielded hyperparameter points that performed better on unseen problems in preliminary experiments than the exhaustive strategy did. Although, for a given number of points, the exhaustive strategy can produce hyperparameter sets of lower cost overall, they do not generalize to unseen problems as well (details in appendix D). Of course the exhaustive procedure is also more expensive compared to the greedy strategy, and the computational expense increases rapidly with the size of hyperparameter set being computed.

# 3 Experiments

In this work our goal is to design hyperparameter lists that generalize to unseen problems. We designed two main experiments to test the ability of our method to produce good hyperparameter lists that perform well (compared to alternative tuning procedures at similar tuning budgets) on workloads not used in their construction: a leave-one-out cross validation experiment at the workload level, and an experiment on a separate "test set" of workload variants, selected by hand from the pool of AlgoPerf workload variants.

## 3.1 Leave-one-out cross-validation experiment

We constructed a simple leave-one-out cross validation test, at the workload level, to test how well the hyperparameter lists we generated performed on workloads that weren't used to construct them. We sampled 200 hyperparameter points from our broad search space and ran each point on the 8 AlgoPerf base workloads, yielding a total of 1600 trials composed of 8 groups of 200 trials. We then used our greedy procedure (Section 2.6) to construct 8 different 5-point hyperparameter lists, each leaving out one of the workloads (i.e. only computing the cost on the subset of trials from the 7 remaining workloads). Then we measured how well each 5-point list performed on the held-out workload compared to various baselines (described below).

As shown in Appendix D an exhaustive strategy was also considered, but it underperformed in such leave-one-out tests, indicating that such a procedure would produce hyperparameter lists that overfit on workloads used to develop the list.

### 3.1.1 Baselines

We constructed several baselines based on alternative procedures for tuning using a small number of tuning trials. Ultimately, a 5-point hyperparameter list is not meant to compete with hundreds or thousands of tuning experiments using progressively more refined search spaces, it merely needs to provide value beyond other procedures that are similarly affordable computationally and require similarly little workload-specific information.

- Learning rate sweeps: We defined a learning rate range of [1e-4, 1e-2] and sampled 5 points on a log scale. The learning rate range matches our broad search space range from Table 1. One question a practitioner might face while running learning rate sweeps is what values to use for regularization hyperparameters such as weight decay, dropout, label smoothing etc. Dahl et al. (2023) showed the importance of tuning regularization hyperparameters to get the best results, but given a limited budget of 5 trials it is hard to explore a wide range of regularization hyperparameter values while also exploring the learning rate. Therefore, we created 2 versions of our learning rate sweep baseline, one with weight decay set to 0.5, and one with weight decay set to 1e-4. In this way one sweep matches the higher end of possible weight decay values from the broad search space and the other matches the lower end.

- Vizier search: We used Vizier (Golovin et al., 2017) search (with default settings) with 5 sequential trials on our broad search space from Table 1 to optimize validation metrics for each of the AlgoPerf base workloads. Using a budget of 5 sequential trials is more generous than using a budget of 5 parallel trials, as sequential trials can leverage results from earlier to make better choices. Asking an off-the-shelf Bayesian optimization tool, such as Vizier, to minimize validation error on a workload isn't quite the same as minimizing training time to achieve a particular validation error threshold, but we can still conclude that searches that fail to achieve the AlgoPerf evaluation metric target for a given workload are not training successfully.

- We used the best proposed 5-point hyperparameter list from (Metz et al., 2020), as is, to evaluate the performance of an ordered hyperparameter list that exists in literature. We replicated the exact update rule used by (Metz et al., 2020) to make sure there were no discrepancies between the definitions of optimization hyperparameters in the training code due to differences in our experimental setup.

### 3.2 Held-out workloads experiment

We tested our final hyperparameter list (derived from trials on all 8 ALGOPERF base workloads) on ALGOPERF workload variants to evaluate if the generated list is robust to architectural modifications designed to require different optimal hyperparameter choices. Some of the ALGOPERF workload variants have targets that are harder to achieve than others, compared to hyperparameter configurations that do well on the corresponding base workload. For example, the IMAGENET VIT POST-LN variant shifts the layer normalization position and makes training with Adam (and related algorithms) somewhat unstable at the beginning, thus potentially necessitating a longer warmup phase or some other mitigation. Hyperparameter lists having a wide range of learning rates, warmup step fractions, and weight decay strengths are required to train successfully on the more challenging ALGOPERF workload variants that we attempted to select (see Section 2.4).

## 4 Results

### 4.1 Our proposed 5-point hyperparameter list achieves a tuning efficiency gain of at least 3x on most AlgoPerf workloads.

We can compare our 5-point hyperparameter lists to simply tuning with 5 trials of random search on the original broad search space, in the leave-one-workload-out cross validation setting. Using the greedy procedure in Section 2.6, we constructed 8 different 5-point hyperparameter lists, each based on holding out a different workload. We used the Opda software package (Lourie et al., 2023) to generate 70 percent confidence bands for the broad search space tuning curves on each workload. Figure 1 plots the broad search space tuning curve (validation performance vs number of trials of random search) alongside the validation error achieved by a 5-point hyperparameter list—*that did not use results on that workload during list construction*—for two representative workloads: one where the 5-point hyperparameter list performs well (IMAGENET RESNET, left), and one where the 5-point hyperparameter list isn't providing a clear tuning efficiency advantage (CRITEO 1TB, right).

For most of the ALGOPERF base workloads, a hyperparameter list of size 5 that did not use results on that workload during list construction is as good as a random search with a tuning budget of at least 15 on the broad search space (see appendix B for results on all workloads).

On CRITEO 1TB DLRM and WMT workloads, unlike most workloads, our hyperparameter lists do not provide a clear tuning efficiency advantage over random search and have similar performance to a random search of the same budget. Our hyperparameter lists of size 5 are never worse than random search of same budget, but the CRITEO 1TB DLRM and WMT workloads seem somewhat unique among the 8 ALGOPERF base workloads in terms of what hyperparameter configurations perform well. If our workload library was larger and and contained other similar workloads (e.g. containing another sequence model trained on natural language text), we would hope that the hyperparameter list construction procedure would be much more likely to select a point that could perform well WMT, showing one path to constructing even more general and useful hyperparameter lists, namely deriving them from a larger and more diverse library of workloads. Note that the final list we recommend is based on all 8 ALGOPERF base workloads and should work well on workloads similar to the WMT workload.

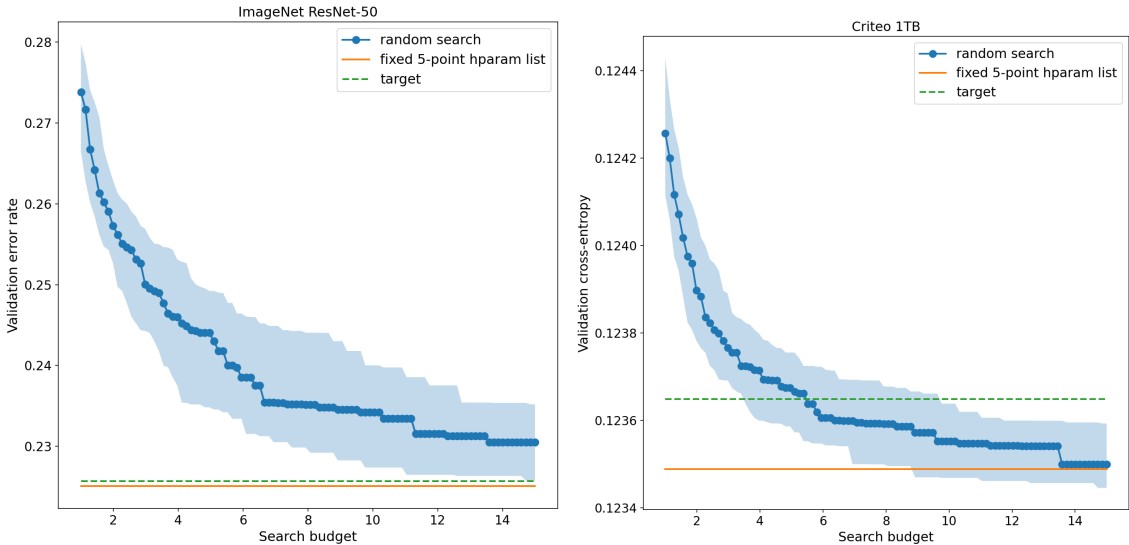

Figure 1: Comparison of different tuning budgets vs proposed 5-point hyperparameter list for IMAGENET RESNET and CRITEO 1TB DLRM workloads. The tuning efficiency gains vary from workload to workload with IMAGENET RESNET showing the highest gains as the 5-point hyperparameter list being as efficiency as broad search with 3x the tuning budget. For our CRITEO 1TB DLRM and WMT workloads the 5-point hyperparameter list is as efficient as random search with same tuning budget. See Appendix B for results on other workloads.

### 4.1.1 Our broad search space contains hyperparameter points that work well on multiple workloads

One essential requirement of our hyperparameter list construction procedure is that the sample from the initial broad search space must contain enough points that perform well on more than one workload. The extent to which hyperparameter configurations exist that transfer across workloads, even in a limited way, is also of independent interest. Table 4 counts how many points from the sample of 200 points from the broad search space described in Section 2.2 train successfully (achieve the ALGOPERF validation error target for the workload) on each workload, as well as how many of those train successfully on an additional one or two other workloads. Points that don't traing successfully on at least two workloads are almost impossible to select with any reasonable hyperparameter list construction procedure when evaluating it in a leave-one-workload-out setting, as we do in this work, since any reasonable procedure should be selecting points that do well according to at least *some* measurement it has access to.

| **Workload** | # points that train successfully | # points that also train successfully on another workload | # points that also train successfully on two other workloads |
|---|---|---|---|
| CRITEO 1TB DLRM | 24 | 14 | 4 |
| FASTMRI | 8 | 3 | 1 |
| IMAGENET RESNET | 2 | 2 | 2 |
| IMAGENET VIT | 17 | 10 | 4 |
| LIBRISPEECH CONFORMER | 6 | 6 | 4 |
| LIBRISPEECH DEEPSPEECH | 23 | 19 | 8 |
| OGBG GNN | 9 | 6 | 2 |
| WMT TRANSFORMER | 7 | 6 | 1 |

Table 4: Number of hyperparameter points that reach the target defining successful training per workload. The second and third columns include the number of points that train successfully on at least one other workload, or at least two other workloads, respectively (among the 8 ALGOPERF base workloads)

### 4.2 Our hyperparameter lists do well on unseen problems in leave-one-out cross-validation tests

As we saw in Section 4.1, on most workloads, we are much better off using a precomputed hyperparameter list than using random search from a broad search space with the same budget, even when the hyperparameter lists are derived exclusively from other workloads. But how well do the hyperparameter lists we generated compare to other alternative tuning procedures that can be applied to a new problem, such as simple learning rate sweeps? Once again, we used 5-point hyperparameter lists in a leave-one-out cross validation setting to evaluate the validation performance on problems that were not used to construct the lists. We compared them to the various baselines described in Section 3.1.1, namely learning rate sweeps with different fixed weight decay values, Vizier using default settings, but with a more generous budget of either 5 or 10 sequential trials, and the hyperparameter list proposed by Metz et al. (2020). To minimize the effect of training outcome variance on our results and to comply with the ALGOPERF benchmark rules, we ran each of the five hyperparameter points five times, resulting in 25 trials, and took the median training time over the five repetitions to report in Table 5 (training times are reported as the fraction of the budget used).

|  | Criteo 1TB | fastMRI | ImageNet | | LibriSpeech | | OGBG | WMT |
|---|---|---|---|---|---|---|---|---|
|  | DLRMsmall | U-Net | ResNet-50 | ViT | Conformer | DeepSpeech | GNN | Transformer |
| Leave-one-out (ours) | 0.73889 | **0.12968** | **0.93967** | **0.92967** | **0.83** | **0.79** | **0.54** | $\infty$ |
| LR sweep (WD = 0.5) | 0.83874 | $\infty$ | $\infty$ | $\infty$ | $\infty$ | $\infty$ | 0.72 | $\infty$ |
| LR sweep (WD = 1e-4) | **0.71892** | $\infty$ | $\infty$ | $\infty$ | $\infty$ | $\infty$ | $\infty$ | 0.69983 |
| Vizier (budget = 5) | $\infty$ | $\infty$ | 0.979653 | $\infty$ | $\infty$ | $\infty$ | $\infty$ | 0.999752 |
| Vizier (budget = 10) | $\infty$ | $\infty$ | 0.979653 | $\infty$ | $\infty$ | $\infty$ | 0.94 | 0.999752 |
| Metz et al | $\infty$ | $\infty$ | $\infty$ | $\infty$ | $\infty$ | $\infty$ | $\infty$ | $\infty$ |

Table 5: Step fractions to hit targets on ALGOPERF base workloads where step fraction is defined as the ratio of step at which target is first reached to the total number of steps for that workload. If target is not reached we note the step fraction as infinity.

We found that the best 5-point hyperparameter list from Metz et al. (2020) doesn't train successfully (achieve the validation error target in time) on any of the base workloads. Perhaps the most plausible reason is the difference in scale between the workloads in ALGOPERF vs the workloads used Metz et al. (2020), but other methodological differences might also play a role.

In addition to training speed measurements in Table 5, we also report the best validation metrics achieved on base workloads for our leave-one-out hyperparameter lists and the other baselines in Table 6. Overall, our hyperparameter lists train more workloads successfully, more quickly, and achieve better validation metrics than the alternative tuning procedures we tried. The learning rate sweep baseline with a small value for weight decay trained the CRITEO 1TB workload the fastest, but it failed to train most other workloads successfully, let alone as quickly as our leave-one-workload-out hyperparameter lists. Vizier search with a 2X larger number of tuning trials and the ability to adaptively respond to earlier trials did produce the best BLEU score on the WMT workload, but even ignoring is budget advantages, that feat was not replicated for any other workloads.

|  | Criteo 1TB | fastMRI | ImageNet | | LibriSpeech | | OGBG | WMT |
|---|---|---|---|---|---|---|---|---|
|  | DLRMsmall | U-Net | ResNet-50 | ViT | Conformer | DeepSpeech | GNN | Transformer |
| Metric | CE $\downarrow$ | SSIM $\uparrow$ | Error Rate $\downarrow$ | Error Rate $\downarrow$ | WER $\downarrow$ | WER $\downarrow$ | mAP $\uparrow$ | BLEU $\uparrow$ |
| Leave-one-out (ours) | **0.123488** | **0.723800** | **0.22508** | **0.22124** | **0.079682** | **0.116066** | **0.289699** | 30.417497 |
| LR sweep (WD = 0.5) | 0.123542 | 0.722731 | 0.22684 | 0.62042 | 0.119027 | 0.124449 | 0.283971 | 24.045937 |
| LR sweep (WD = 1e-4) | 0.123521 | 0.722462 | 0.25078 | 0.25462 | 0.106376 | 0.125401 | 0.275011 | 30.855032 |
| Vizier (budget = 5) | 0.123861 | 0.711889 | 0.22672 | 0.24966 | 0.084357 | 0.124603 | 0.280878 | 30.755536 |
| Vizier (budget = 10) | 0.123861 | 0.711889 | 0.22672 | 0.22956 | 0.084357 | 0.118377 | 0.280878 | **30.855786** |
| Metz et al | 0.123719 | 0.722440 | 0.27878 | 0.24284 | 0.114179 | 0.123489 | 0.258499 | 28.466302 |
| Random Search (budget = 5) | 0.123975 | 0.721746 | 0.260200 | 0.259580 | 0.122765 | 0.136830 | 0.257632 | 29.46569 |

Table 6: Best validation metrics achieved on ALGOPERF base workloads.

### 4.2.1 As hyperparameter list size increases, our cost function ($\mathcal{C}$) decreases then saturates

Figure 2 shows how the length of the hyperparameter lists in the leave-one-out setup affect the number of workloads trained successfully and the average cost over left-out workloads. At 7 point lists, every base workload can be trained successfully. At 7 point lists, the average cost across left-out workloads has also flattened. Although additional hyperparameter points beyond 7 does not improve the cost on the ALGOPERF base workloads, on a larger set of workloads (or on a novel workload) it is possible the additional points would add value.

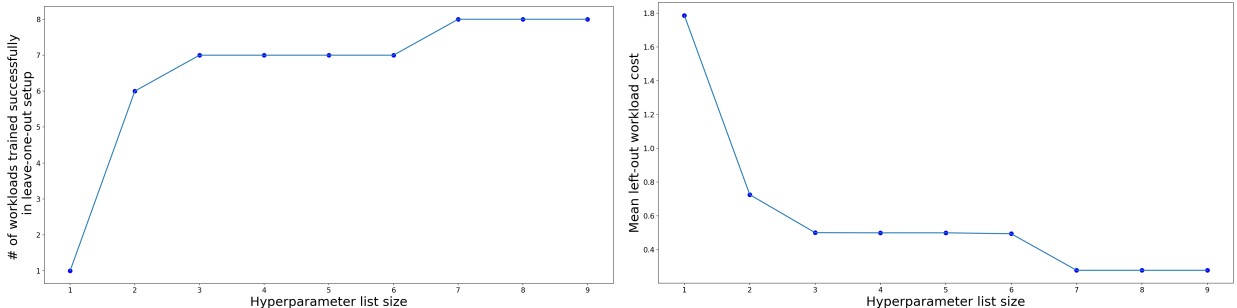

Figure 2: left: number of left-out workloads trained successfully vs hyperparameter list size, right: mean $\mathcal{C}$ over left-out workloads vs hyperparameter list size.

### 4.3 Our proposed 5-point hyperparameter list is robust to architectural modifications.

In addition to studying how well the hyperparameter lists we constructed perform in the leave-one-workload-out setting, we can use the workload variants we selected in Section 2.4 to see if they are robust to minor workload modifications designed to change the optimal hyperparameters somewhat. We constructed a 5-point hyperparameter list as detailed in Section 2.6 by applying our greedy procedure over trials from *all* 8 Algoperf base workloads. We then evaluated this final 5-point hyperparameter list on the workload variants we selected. Table 7 shows the fraction of the budget required to train successfully on each workload variant, using points from the hyperparameter list we constructed. The final hyperparameter list trains successfully on 7 out 8 of the variants, demonstrating that the list as a whole is relatively robust to these kinds of architectural modifications, even though they can change the optimal hyperparameters. Table 8 shows the corresponding best achieved validation metrics on workload variants. Although the VIT + POST-LN variant was not trained successfully, our final hyperparameter list still had a point in it that reached a reasonably good validation error rate. For completeness, Table 9 also reports the step fractions for our final (using all the base workloads) 5-point hyperparameter list on ALGOPERF base workloads. These results *could* be slightly optimistic because the base workloads were used in constructing the final list, so for most purposes the leave-one-workload-out results will be more realistic. However, comparing the leave-one-out cross validation results in the first row of Table 5 to Table 9 does not show the base workload results strictly dominating, indicating that with only 5 points there does not seem to be very strong evidence of overfitting to the workload library.

| | Criteo 1TB | fastMRI | ImageNet | | LibriSpeech | | OGBG | WMT |
|---|---|---|---|---|---|---|---|---|
| | DLRMSMALL + PRE-LN | U-NET + PRE-LN | RESNET-50 + SiLU | VIT + POST-LN | CONFORMER + GELU | DEEPSPEECH + TanH | GNN + GELU | TRANSFORMER + POST-LN |
| Final 5-point hyperparameter list | 0.71892 | 0.119705 | 0.839703 | $\infty$ | 0.72 | 0.7 | 0.3 | 0.619847 |

Table 7: Step fractions to hit targets on Algoperf heldout workloads where step fraction is defined as the ratio of step at which target is first reached to the total number of steps for that workload. If target is not reached we note the step fraction as infinity.

| | Criteo 1TB | fastMRI | ImageNet | | LibriSpeech | | OGBG | WMT |
|---|---|---|---|---|---|---|---|---|
| | DLRMSMALL + PRE-LN | U-NET + PRE-LN | RESNET-50 + SiLU | ViT + POST-LN | CONFORMER + GELU | DEEPSPEECH + TanH | GNN + GELU | TRANSFORMER + POST-LN |
| Metric | CE ↓ | SSIM ↑ | Error Rate ↓ | Error Rate ↓ | WER ↓ | WER ↓ | mAP ↑ | BLEU ↑ |
| Final 5-point hyperparameter list | 0.123536 | 0.724563 | 0.2245 | 0.26942 | 0.083247 | 0.135851 | 0.284199 | 30.54408 |

Table 8: Best validation metrics achieved by our proposed 5-point hyperparameter list on Algoperf workload variants. Each metric is reported as median of 5 trials to reduce variance.

| | Criteo 1TB | fastMRI | ImageNet | | LibriSpeech | | OGBG | WMT |
|---|---|---|---|---|---|---|---|---|
| | DLRMSMALL | U-NET | RESNET-50 | ViT | CONFORMER | DEEPSPEECH | GNN | TRANSFORMER |
| Final 5-point hyperparameter list | 0.75886 | ∞ | 0.949664 | 0.809714 | 0.82 | 0.79 | 0.58 | 0.659837 |

Table 9: Step fractions to hit targets on ALGOPERF base workloads where step fraction is defined as the ratio of step at which target is first reached to the total number of steps for that workload. If target is not reached we note the step fraction as infinity.

## 4.4 Final 5-point hyperparameter list

Table 10 provides the details of our hyperparameter points for the final list that practitioners can try on their own problems. Note that we used a learning rate schedule consisting of a warmup phase followed by a cosine decay to zero. The warmup fraction in a particular hyperparameter point is the length of the warmup portion used for ALGOPERF workloads in terms of the total training step budgets. In other words, a warmup fraction of 0.02 indicates 2% of the entire training run was used for learning rate warmup. Our warmup starts at zero and ends at the Base Learning Rate (Base LR) values indicated in Table 10 (which is then followed by cosine decay all the way to zero by the end of the training runs). Due to the use of a non-constant learning rate schedule, in order to apply our final hyperparameter list to a new problem, it is important to have some estimate of the total number of training steps. When interpreting the optimization hyperparameters, note that the NadamW update rule that we used (provided in Optax) uses bias correction.[5]

| Id | Base LR | Warmup Fraction | $\beta_1$ | $\beta_2$ | Weight Decay | Dropout Rate | Label Smoothing |
|---|---|---|---|---|---|---|---|
| 1 | 0.007188680089024849 | 0.1 | 0.9521079797438937 | 0.9545645606521953 | 0.020932289532959312 | 0.0 | 0.2 |
| 2 | 0.0011719210768906827 | 0.02 | 0.9641782560318817 | 0.9953311727740848 | 0.15957548811577366 | 0.1 | 0.0 |
| 3 | 0.001183374563441696 | 0.02 | 0.918959806679234 | 0.9941923836947718 | 0.028400661323288435 | 0.1 | 0.1 |
| 4 | 0.0014515212275017363 | 0.1 | 0.9600296609757403 | 0.889423091749684 | 0.031808785805059143 | 0.0 | 0.2 |
| 5 | 0.0005102205206215031 | 0.05 | 0.9120180064671332 | 0.9597041640569521 | 0.04833675039698776 | 0.1 | 0.0 |

Table 10: Final hyperparameter points in priority order.

As mentioned earlier, the hyperparameter points include regularization hyperparameter values, as we found that it makes a difference in the final validation performance achieved; it's important to carefully set dropout and label smoothing values, where applicable. The final 5-point hyperparameter list provides coverage for a wide range of base learning rates, spanning multiple orders of magnitude, as some problems require smaller learning rates and others require larger learning rates.

---

[5]https://github.com/google-deepmind/optax/blob/main/optax/_src/transform.py#L319C15-L319C35

## 5 Discussion

The primary use case for our proposed hyperparameter list is as a strong first baseline result on a new workload in resource-constrained settings. An important hyperparameter that is part of our prescription is the cosine-decay learning-rate schedule which requires fixing a choice for the training horizon in terms of the number of steps before training. For an entirely new problem we may not have complete knowledge of such a horizon and it's left to the practitioner's discretion to select a step budget via trial and error. In this work our hyperparameter lists were derived on ALGOPERF: Training Algorithms benchmark problems, which happen to have well-specified step budgets, and thus it's unclear what step budget to use when faced with a completely new problem. For problems very similar to ALGOPERF workloads practitioners could use the same step budgets, but in general knowing how long to train (or what validation error rate is realistic to expect) is quite difficult on a truly novel workload.

For a practitioner who has developed a new training algorithm, we suggest our hyperparameter list as a strong baseline to compare to. Furthermore, we suggest the procedure employed in the paper to generate hyperparameter lists as a candidate procedure towards identifying good hyperparameter combinations for the algorithm. In particular we stress on the leave-one-workload out testing for hyperparameter settings, to identify the robustness of proposed hyperparameter settings.

We note that we have not tested our hyperparameter points on large scale problems like many of the LLM workloads prevalent in the industry with billions of parameters but we do think our hyperparameter points could be a good starting point for the scaling ladders typically developed for such problems.

### 5.1 Future Work

An interesting extension of our work is to propose scaling curves for each hyperparameter starting at hyperparameter values suggested in this work to provide a solution that scales well with model size. We also believe that deriving such precomputed hyperparameter lists using the methodology in our work for a new training algorithm would make it easier for the practitioner to adopt such methods. Finally, to address the problem of knowing the training horizon in advance one promising direction is to combine our work with methods that eliminate the need for learning rate tuning as done in Defazio et al. (2024).

## 6 Conclusion

We propose hyperparameter lists of increasing sizes that practitioners can deploy in resource-constrained situations on a new problem if the training horizon is known in advance. Our hyperparameter list performs well on a wide variety of architectures and datasets relevant to practitioners and is robust to architectural modifications. Our hyperparameter list can serve as a strong baseline for the broader research community to look for better solutions in the low resource tuning regime. We encourage the broader community to adopt our methodology to test the generalization capabilities of hyperparameter lists using leave-one-out cross-validation on ALGOPERF benchmark.

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

## A    Algoperf workload implementation details

We report the number of steps used for each of the ALGOPERF workloads used in our experiments below, note that heldout workloads use same number of steps as the corresponding base workload.

| | Criteo 1TB | fastMRI | ImageNet | | LibriSpeech | | OGBG | WMT |
|---|---|---|---|---|---|---|---|---|
| | DLRMSMALL | U-NET | RESNET-50 | VIT | CONFORMER | DEEPSPEECH | GNN | TRANSFORMER |
| Number of training steps | 10666 | 36189 | 186666 | 186666 | 80000 | 48000 | 80000 | 133333 |

Table 11: Number of steps used for training ALGOPERF workloads.

All our experiments were run using code from init2winit github repository (Gilmer et al., 2023) on TPUv2 2x2 slices (16GB HBM per chip), except the CRITEO 1TB workload which was trained on TPUv3 4x4 (32GB HBM per chip) due to higher memory requirements in the embedding layers.

## B    Broad search space vs hyperparameter list

We include comparison of tuning curves from the broad search space vs our proposed hyperparameter list for all ALGOPERF base workloads.

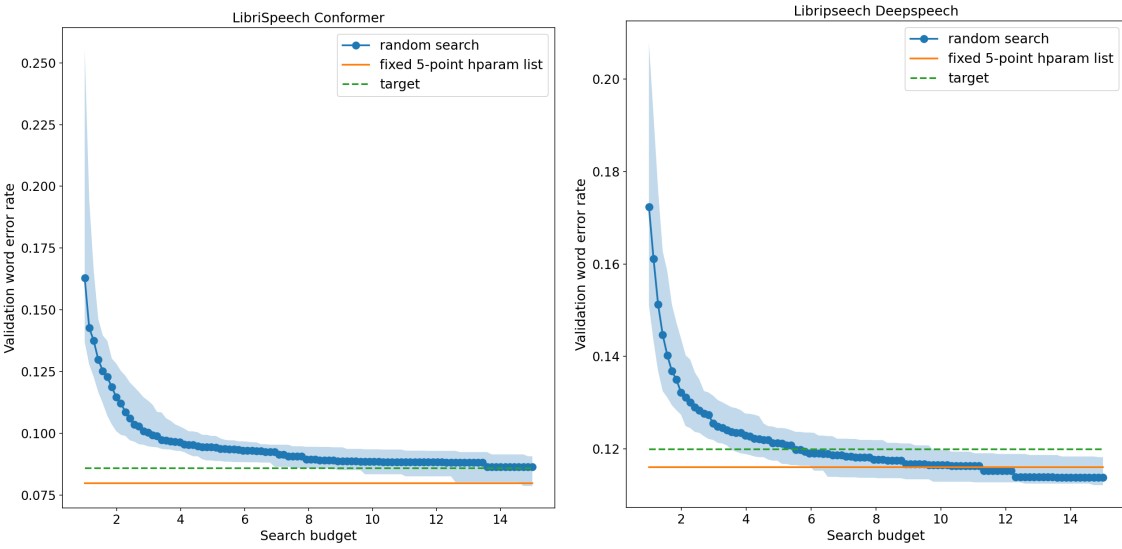

Figure 3: Comparison of different tuning budgets vs proposed 5-point hyperparameter list for LIBRISPEECH CONFORMER and LIBRISPEECH DEEPSPEECH workloads

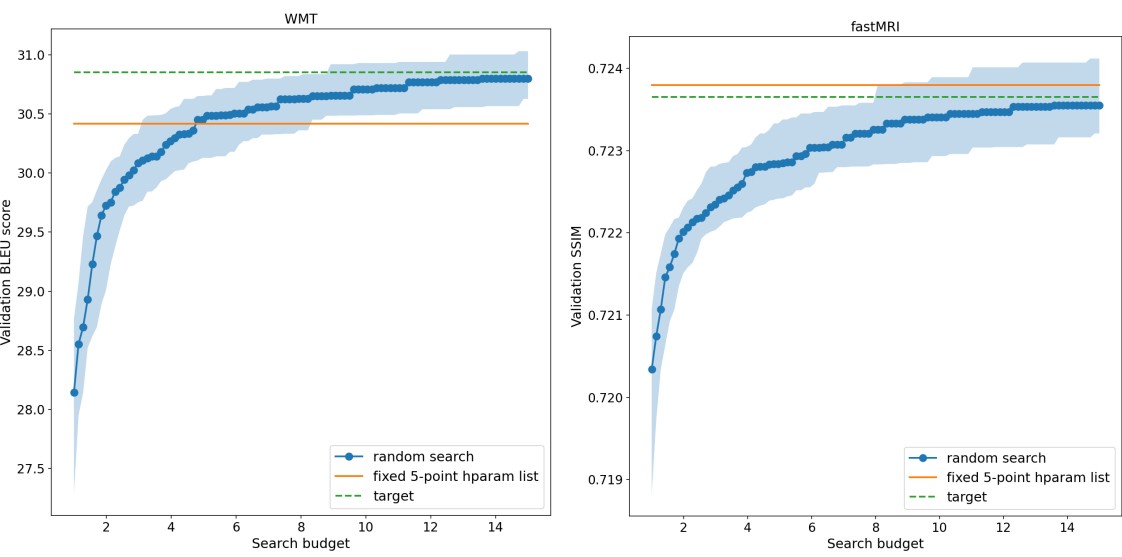

Figure 4: Comparison of different tuning budgets vs proposed 5-point hyperparameter list for WMT TRANS-FORMER and CRITEO 1TB DLRMSMALL workloads

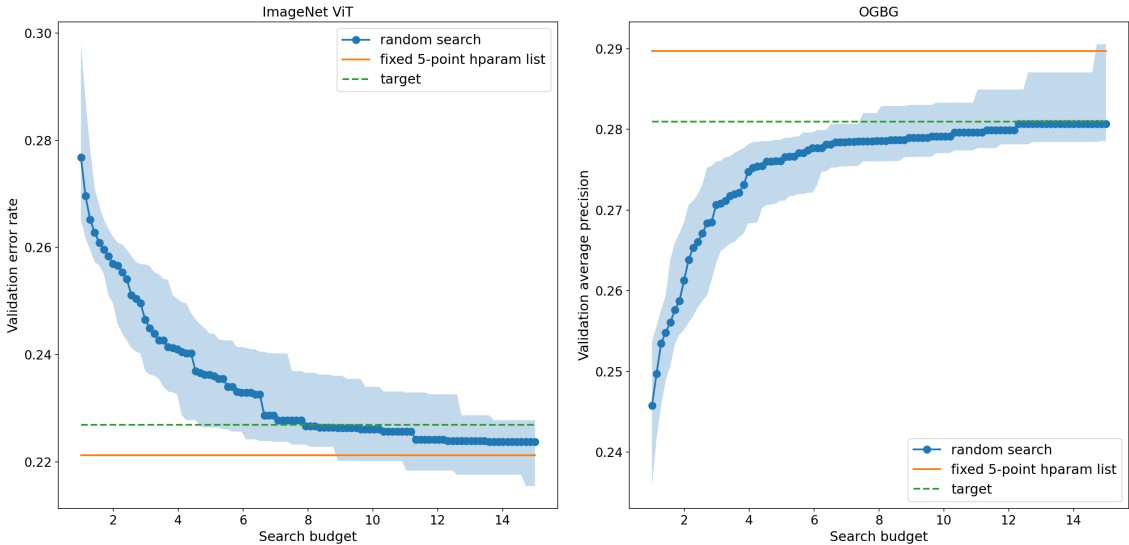

Figure 5: Comparison of different tuning budgets vs proposed 5-point hyperparameter list for IMAGENET VIT and OGBG GNN workloads

## C    Choice of penalty factor in Cost function to evaluate hyperparameter lists

We discuss how choosing a penalty factor of 1.0 overfits to choices that do no generalize on unseen problems. We vary the penalty factor from 1.0 to 2.0 at a fixed desired hyperparameter list size of 5 and observe that penalty factors >= 1.33 lead to hyperparameter lists that do better on unseen problems in leave-one-out setup.

| Penalty factor | # of left-out workloads trained successfully |
|---|---|
| 1.0 | 6 |
| 1.11 | 6 |
| 1.22 | 6 |
| 1.33 | 7 |
| 1.44 | 7 |
| 1.55 | 7 |
| 1.66 | 7 |
| 1.77 | 7 |
| 1.88 | 7 |
| 2.0 | 7 |

Table 12: Number of targets hit in leave-one-out setup on ALGOPERF base workloads.

## D    Exhaustive vs Greedy procedure to pick hyperparameter list candidates

We show that the exhaustive procedure gets us hyperparameter lists that have lower cost but tend to overfit to input workloads and do not hit targets on unseen problems as seen below :

| Strategy (list size) | # of left-out workloads trained successfully |
|---|---|
| greedy (4) | 7 |
| exhaustive (4) | 5 |

Table 13: Number of targets hit in leave-one-out setup on ALGOPERF base workloads.

| | Criteo 1TB | fastMRI | ImageNet | | LibriSpeech | | OGBG | WMT |
|---|---|---|---|---|---|---|---|---|
| | DLRMSMALL | U-NET | RESNET-50 | VIT | CONFORMER | DEEPSPEECH | GNN | TRANSFORMER |
| Final 5-point hyperparameter list | 0.75886 | ∞ | 0.949664 | 0.809714 | 0.82 | 0.79 | 0.58 | 0.659837 |

Table 14: Step fractions to hit targets on Algoperf base workloads where step fraction is defined as the ratio of step at which target is first reached to the total number of steps for that workload. If target is not reached we note the step fraction as infinity.

