# OpenReview forum: "Training neural networks faster with minimal tuning usingpre-computed lists of hyperparameters for NAdamW"
_TMLR — Rejected by TMLR_

### Review · Reviewer_1M7Y · 2024-12-12

**Summary Of Contributions:**

The authors present a method for hyperparameter tuning. Divide a library of workloads into training and holdout, and  use a greedy search algorithm to find an ordered list of hyperparameter combinations (from a potentially broad search space) that works well on the training set of workloads, as in for each workload, one combination from the list results in good performance. The authors show that the list obtained is useful for hyperparameter tuning on holdout workloads: trying hyperparameters from this list is often more efficient than a quasi-random search.

The idea of a pre-selected list has been presented by Metz et al., as the authors cited. The main difference of this work is that this algorithm works well in more realistic settings, namely on the AlgoPerf benchmarks.

**Audience:**

Yes

**Broader Impact Concerns:**

No concerns.

**Claims And Evidence:**

Yes

**Requested Changes:**

It would strengthen the paper if the cost of producing the hyperparameter list is considered when accounting for the time/compute cost of the proposed method. If the authors are optimistic that a pre-selected list would have sufficiently broad application such that the upfront cost is negligible per use, evidence in that direction could be presented.

**Strengths And Weaknesses:**

- The paper provides a list of hyperparameters that can be used off-the-shelf for many popular workloads. This contribution would be quite useful for academics looking for a reasonable start to train their models.

- The paper devices a cost function based on training time that can evaluate the performance of a meta learning algorithm over disparate workloads. This cost function could see some applications beyond the selection of learning rates etc.

- The experimental results provide convincing evidence that their hyperparameter list outperforms a number of baselines, using leave-one-out and hold out performance.

- However, with the exception of Metz et al., these baseline are limited to methods that do not have access to similar workloads prior to tuning on the target workloads. The advantage could be less pronounced when the cost of upfront investments (i.e. the compute that goes into producing the list) is factored into the equation. In order to perform well on a new workload, the hyperparemeter list must have seen some relevant trials in its training, making it difficult for the list to be broadly useful without incurring a large compute burden upfront. If the hyperparameter list needs to be obtained by training over workloads of similar scale as the target workloads, it could be significantly more costly than e.g. applying the scaling laws.

- While the performance of the methodology is good overall, its performance is not quite predictable on individual workloads, especially one that deviates from the training set. As such the applications of the hyper parameter list may be limited.

---

> ### Author Response · Authors · 2025-01-22
>
> **Note**: We apologize for missing the author response deadline. Although a decision has already been entered for our submission, we've decided to add brief replies to a subset of reviewer points anyway. Obviously we don't expect a response from any reviewers at this stage, although we would welcome one if any reviewers are so inclined.
>
> Thanks for reviewing our paper and suggesting edits that make our paper stronger. Adding the cost of computing our hyperparameter list is a great suggestion. Our claim is that our hyperparameter lists generalize *better* than baselines we compared to (such as learning rate sweeps) in a leave-one-workload-out cross validation setting, under the constraint of only using 5 trials. The more workloads we can use to create the hyperparameter list, the better it’d work on unseen problems. Given the performance in our leave-one-out tests and on the separate workload variants test, we believe our hyperparameter list should do better than the typical simple manual sweeps researchers end up doing when they are budget-constrained.

---

### Review · Reviewer_rjkg · 2024-12-20

**Summary Of Contributions:**

The paper explores methods to identify a list of robust hyperparameters that generalize across multiple workloads. This list is particularly useful in resource-constrained scenarios where conventional tuning protocols may yield suboptimal results. The authors propose a straightforward greedy algorithm that iteratively selects hyperparameter configurations, prioritizing those that reduce the overall cost of the list the most.

**Audience:**

Yes

**Claims And Evidence:**

No

**Requested Changes:**

See Strengths And Weaknesses.

**Strengths And Weaknesses:**

**Strengths**
- The simplicity of the proposed method makes it accessible and easy to use for parctitioners.
- Focusing on the AlgoPerf benchmark is good, as it encompasses diverse modalities, enhancing the method's applicability.

**Weaknesses**
- **Hard to see the novelty in Methods section:** The Methods section should clearly distinguish between the proposed method and the AlgoPerf benchmark. Right now, it's hard to tell where the explanation of the new method begins and what the starting point is. Consider creating a separate section for the AlgoPerf benchmark to keep it distinct from the method. This will make it easier to highlight the novelty of your approach. Additionally, clearly stating the assumptions in the AlgoPerf section would help improve the paper’s structure and readability.
- **Wording can be improved:** The wording can be refined to make the discussions easier to follow. For instance, the term workload is introduced in Section 2.3 but is used extensively before that. This makes it unclear what workload refers to until it is defined. I suggest defining key terms like workload earlier in the text to enhance clarity and readability.
- **Simplistic Baselines:** I am not an expert in the field, but for me the baselines seem really simplistic. I expected to see more state-of-the-art strategies included for a more robust comparison. Currently, only one method from the literature is used as a baseline, making it difficult to evaluate the effectiveness of the proposed method. While the authors mention that heuristic-based approaches are orthogonal to their work, it is unclear why they do not include such comparisons. If I have misunderstood something here, I am open to reconsidering my stance.
- **Independence of Workloads and Impact:**  Since the hyperparameter lists are derived from the AlgoPerf Benchmark, it seems problematic that the tested workloads are merely variants rather than truly independent (if I understood correctly). The proposed method could have a much greater impact if it were tested on (more) independent workloads where the dataset, architecture, and training objectives differ. This would provide a more comprehensive evaluation of its generalization capabilities. By focusing on AlgoPerf variants, there is a risk that the method prioritizes robustness to minor modifications rather than genuine generalization to new tasks. If I have misunderstood this aspect, I am open to revisiting my opinion.

Some Minor Problems:
- Table Captions should be above tables.
- AlgoPerf sometime not written in \textsc

---

> ### Author Response · Authors · 2025-01-22
>
> **Note**: We apologize for missing the author response deadline. Although a decision has already been entered for our submission, we've decided to add brief replies to a subset of reviewer points anyway. Obviously we don't expect a response from any reviewers at this stage, although we would welcome one if any reviewers are so inclined.
>
> Thanks for reviewing our paper and suggesting edits that make our paper stronger.
>
> Re: "Simplistic Baselines"
>
> We use baselines that a deep learning practitioner with limited access to compute would typically deploy. When only five (parallel) tuning trials are available, off-the-shelf Bayesian optimization tools do not work well (as shown in our results, since we try such a tool in even more favorable conditions).
>
> Re: “Independence of Workloads and Impact”
>
> Based on the remark "Since the hyperparameter lists are derived from the AlgoPerf Benchmark, it seems problematic that the tested workloads are merely variants rather than truly independent (if I understood correctly)," we think there might be a slight misunderstanding.
>
> We conduct leave-one-workload-out cross validation experiments to avoid the risks the reviewer mentions. In these experiments, our hyperparameter list performs better than the typical learning rate sweeps people do in practice. It is true that we ALSO conduct experiments using variants of AlgoPerf workloads, but we view the leave-one-workload-out experiments using the base workloads as stronger evidence for our claims since the variants are closely related to the corresponding base workloads (although the variants were designed to require somewhat different hyperparameters, the process the original AlgoPerf authors used is not without various issues).
>
> The leave-one-out cross validation setting at the workload level is a better test of whether our hyperparameter lists outperform simple learning rate sweeps than just looking at some single workload. Given a set of workloads, we can split them into a "training" set where we search for good hyperparameter values and a "test" set where we measure how well the hyperparameters transfer, but when we only have a small number of workloads, we get a clearer picture from performing cross validation, just as we benefit from cross validation when dealing with small datasets and training models. The AlgoPerf base workloads are just a convenient pool of diverse deep learning workloads that have clearly established error rate goal values.
>
> Finally, at the risk of belaboring the point, we chose the AlgoPerf benchmark for our experiments because it truly does have reasonably diverse combinations of model architectures, datasets, and training objectives. It covers machine-translation encoder-decoder models using cross-entropy loss, speech recognition models with both attention mechanisms and LSTMs using a CTC loss, graph neural networks, medical image reconstruction with UNets using L1 loss, fully-connected CTR models using large embedding layers and performing binary classification, and multi-class image classifiers of both the ResNet and VIT variety.

---

### Review · Reviewer_PCnk · 2024-12-21

**Summary Of Contributions:**

The paper contributes methods to find hyperparams lists for NAdamW from experiments on the AlgoPerf Benchmark. These hyperparam lists perform well on held out AlgoPerf workloads. Additionally, these hyperparams outperforms basic LR/WD sweeps and one off-the-shelf Bayesian optimization tool which are restricted to the same budget

**Audience:**

No

**Claims And Evidence:**

No

**Requested Changes:**

# Major [Acceptance Critical]
- *Comparison against AlgoPerf baselines.* - Regardless of how the authors describe their contributions, the paper would benefit from absolute metrics of comparison against the results of AlgoPerf Baseline AdamW hyperparameter set. Currently, I have been unable to find any. In particular, I would like to see the authors run AlgoPerf Baselines on consistent hardware, and to analyze performance results of their method against this standard.
- *Clarify Contributions* - To demonstrate an benefit of the method over existing work, the authors need to either 1) demonstrate the strength of this method on a varied set of different dataset and quantity the amount of compute saved, or 2) to display performance above the existing AlgoPerf baseline values. *To reiterate, if a practitioner intends to use this method for experimenting on workloads in or similar to the AlgoPerf baseline, would it not suffice to use the existing baseline values resulting from more extensive sweeps?*
  - In the first case, I would either like to see the generalization of these swept values to datasets separate from Algoperf (i.e. - distinct from the base or variant workloads) through further experimentation and analysis.
  - In the second case, I would like a quantitative comparison on additional workloads using hyperparameters provided by this method vs sweep times from scratch. If no sweep is expected for further use, then it would instead be sufficient to include comparisons of this method to a variety of hyperparameters used in baselines of peer-reviewed papers.
- *Generalization Claims* - I would also like to see the paper to explicitly address that it is unable to locate one hyperparam set which can hit the target on all workloads, and explore claims of generalization.

# Minor
- Mention your weight decay sweep in Section 2.2 - Broad Search Space

**Strengths And Weaknesses:**

# Strengths
* This list benefits those who are newer to ML and who have lower compute capacities. It is an interesting algorithm which samples a hyperparameter space in an interesting way to produce candidates which cover a broad range of hyperparameters.
* The method itself has promise and could save quite a bit of time and compute.


# Weaknesses
1. What is the benefit?
  - If a practitioner intends to use this method for experimenting on workloads in or similar to the AlgoPerf baseline, would it not suffice to use the existing AlgoPerf baseline values resulting from more extensive sweeps? To demonstrate an benefit of the method over existing work, the authors need to either A) display performance above the existing AlgoPerf baseline values or B) demonstrate the strength of this method on a varied set of different dataset and quantity the amount of compute saved. Additional experiments are needed outside of the AlgoPerf Benchmark set to show generalization beyond this dataset.

2. More experiments needed for generalization?
  * Further, I’m concerned about generalization claims when the hyperparameter set was not able to hit the AlgoPerf target for a number of workloads. The workloads and targets of AlgoPerf were chosen as a representative set of tasks which one algorithm or set of algorithms could perform well on and generalize to performance across machine learning tasks. While it could be understandable that the target was not reached on a variant workload such as “ViT+Post-LN”, it is more concerning that a hyperparameter set cannot reach the target on the fastMRI base workload.

3. Why use AlgoPerf metrics?
  * The AlgoPerf competition metrics are to determine fastest training to a certain target, rather than most optimal training over a range of time. The target audience seems to be a practitioner lacking compute for a hyperparam sweep who would also like to locate hyperparam values to train quickly.
  * Would a practitioner not rather use a hyperparameter baseline which returns the best final trained model, rather than the fastest trained model?

---

> ### Author Response · Authors · 2025-01-22
>
> **Note**: We apologize for missing the author response deadline. Although a decision has already been entered for our submission, we've decided to add brief replies to a subset of reviewer points anyway. Obviously we don't expect a response from any reviewers at this stage, although we would welcome one if any reviewers are so inclined.
>
> Thanks for reviewing our paper and suggesting additional experiments to make our contributions stronger.
>
> 1. Regarding “More experiments needed for generalization?”:
>
> Our claim is that our hyperparameter lists generalize *better* than baselines we compared to (such as learning rate sweeps) in a leave-one-workload-out cross validation setting, under the constraint of only using 5 trials. We aren't claiming that in some absolute sense our hyperparameter lists generalize in all cases and conditions, merely that—compared to other practical alternatives—they work well. The leave-one-workload-out setting is the most realistic evaluation protocol we could come up with, and the baselines we constructed are typical approaches a practitioner would try if they’re compute constrained. So indeed while it is true that occasionally our hyperparameter lists don't achieve the goal error rate on ALL workloads (as you point out with FastMRI), it *is* true that averaged over all choices of which workload to leave out, our hyperparameter lists convincingly outperform competing recipes.
>
>
> 2. Regarding the suggestion to "use the existing AlgoPerf baseline values resulting from more extensive sweeps" we weren't sure if that remark referred to the 5-point hyperparameter lists used as prize qualification baselines in the recent AlgoPerf competition (https://github.com/mlcommons/algorithmic-efficiency/blob/main/prize_qualification_baselines/external_tuning/tuning_search_space.json) or whether it referred to the 20-point hyperparameter lists used as baselines in the original AlgoPerf paper (https://arxiv.org/abs/2306.07179). If the former, that baseline was constructed using ALL AlgoPerf workloads and has an advantage over hyperparameter lists constructed in the leave-one-workload-out setting. If the latter, 20 hyperparameter points is way too strong when there are only 8 workloads and is more akin to an "oracle" baseline, and once again it isn't clear how to compare in the leave-one-workload-out setting since that baseline was informed by results on all base workloads.
>
>
> 3. Regarding why we used the AlgoPerf metrics:
>
> We don't know how to aggregate validation error fairly across different workloads with different loss functions and evaluation metrics. By using training time to reach a specified evaluation metric goal value, as AlgoPerf suggests, it is much easier to aggregate across the different types of workloads.
>
> The AlgoPerf targets are set competitively using extensive searches across optimizers and hyperparameters, so reaching these targets indicates good model performance. If one method reaches the target, but another doesn’t (as is common in our results), then the first method is producing a better trained model. However, if two methods both achieve the target on a given workload, the faster one is preferred.

---

### Decision · Action_Editor_k33Z · 2025-01-20

**Recommendation:** Reject

**Comment:**

The paper presents a method for generating pre-computed hyperparameter lists for NAdamW, with mixed reviewer recommendations (1 leaning accept, 2 reject):

- Key Strengths:

1. Practical value for resource-constrained settings
2. Simple, accessible methodology
3. Evaluation on AlgoPerf benchmark
4. Performance advantages over basic tuning approaches

- Major Concerns:

- Limited evidence beyond AlgoPerf benchmark
- Inability to hit targets on some workloads
- Unpredictable performance on deviating workloads

- Unclear differentiation from AlgoPerf baseline values
- Limited comparison to state-of-the-art methods
- Missing analysis of upfront computational costs

- Need for clearer distinction between proposed method and AlgoPerf benchmark
- Lack of truly independent test workloads
- Simplistic baseline comparisons

Given TMLR's emphasis on thorough empirical validation and practical impact, the current version has significant limitations that need addressing. While the work shows promise for practical applications, the lack of evidence for generalization beyond AlgoPerf and insufficient comparison to existing baselines are critical issues.

Recommendation: Reject in current form, with encouragement to resubmit after:

1. Adding experiments beyond AlgoPerf benchmark
2. Including comparison to AlgoPerf baseline hyperparameters
3. Analyzing upfront computational costs
4. Strengthening generalization claims with independent workloads
5. Expanding baseline comparisons

**Audience:**

Yes, this work would interest TMLR's audience, particularly:

- Practitioners with limited computational resources for hyperparameter tuning
- Researchers working on automated optimization methods
- Those seeking practical, off-the-shelf hyperparameter settings for common workloads
- Academic users looking for reasonable starting points for model training

However, one reviewer questioned the audience fit due to limited generalization evidence.

**Claims And Evidence:**

The paper's claims about hyperparameter lists for NAdamW are partially supported by evidence, with some significant limitations:

*Strengths:*

- Comprehensive evaluation on AlgoPerf benchmark workloads
- Clear demonstration of performance advantages over basic learning rate/weight decay sweeps
- Well-documented experimental methodology

*Limitations:*

- Insufficient evidence of generalization beyond AlgoPerf benchmark
- Lack of comparison against AlgoPerf baseline hyperparameters
- Mixed results when workloads deviate significantly from training set
- Missing analysis of upfront computational costs for creating hyperparameter lists

**Resubmission Of Major Revision:**

The authors may consider submitting a major revision at a later time.